# Construction of hierarchically porous metal–organic frameworks through linker labilization

Shuai Yuan[1], Lanfang Zou[1], Jun-Sheng Qin[1,2], Jialuo Li[1], Lan Huang[3], Liang Feng[1], Xuan Wang[1], Mathieu Bosch[1], Ali Alsalme[2], Tahir Cagin[3,4] & Hong-Cai Zhou[1,2,3]

A major goal of metal–organic framework (MOF) research is the expansion of pore size and volume. Although many approaches have been attempted to increase the pore size of MOF materials, it is still a challenge to construct MOFs with precisely customized pore apertures for specific applications. Herein, we present a new method, namely linker labilization, to increase the MOF porosity and pore size, giving rise to hierarchical-pore architectures. Microporous MOFs with robust metal nodes and pro-labile linkers were initially synthesized. The mesopores were subsequently created as crystal defects through the splitting of a pro-labile-linker and the removal of the linker fragments by acid treatment. We demonstrate that linker labilization method can create controllable hierarchical porous structures in stable MOFs, which facilitates the diffusion and adsorption process of guest molecules to improve the performances of MOFs in adsorption and catalysis.

[1] Department of Chemistry, Texas A&M University, 3255 TAMU, College Station, Texas 77843-3255, USA. [2] Chemistry Department, College of Science, King Saud University, Riyadh 11451, Saudi Arabia. [3] Department of Materials Science and Engineering, Texas A&M University, College Station, Texas 77843-3003, USA. [4] Artie McFerrin Department of Chemical Engineering, Texas A&M University, College Station, Texas 77843-3022, USA. Correspondence and requests for materials should be addressed to H.-C.Z. (email: zhou@chem.tamu.edu).

Metal–organic frameworks (MOFs), also known as porous coordination networks (PCNs), have attracted broad research interests in the past decades, owing to their wide range of potential applications including gas storage, separation and catalysis[1–3]. Compared with traditional porous solids, such as zeolites, activated carbon and mesoporous silica, MOFs allow for precise design of framework structures and tailoring of pore environments at the molecular or atomic level[4,5]. The pore sizes of most reported MOFs are limited to micropore range (pore diameter smaller than 2 nm), which is suitable for the adsorption and separation of small guest molecules, such as gas, small organic molecules and coordination complexes. Larger biomolecules, however, are usually excluded from MOF cavities, limiting their application. Therefore, the augmentation of MOF pore size to mesoporous range (2 to 50 nm) is highly desired, yet still challenging to date.

In principle, the pore size of a MOF material can be increased through isoreticular expansion with elongated organic linkers[6–8]. One of the representative examples is the successful expansion of MOF-74 into a series of isoreticular structures with pore apertures ranging from 1.4 to 9.8 nm (ref. 9). However, this method only proves to be effective for certain MOF systems, whereas in most cases, elongated linkers often result in interpenetrated structures[10] or undesired topologies[11], which restricts the pore size and porosity. In addition, the elongation of linkers usually weakens the framework stability, leading to fragile MOFs that easily collapse upon the removal of guests[12]. Alternatively, large pores can be fabricated into MOF materials as crystal defects, creating hierarchical-pore architectures[13–16]. Two common approaches have been developed to introduce hierarchical pores to microporous MOFs, including (a) perturbation-assisted synthesis[17,18] and (b) template directed synthesis[19,20] The perturbation-assisted synthesis is carried out under strong stirring, which kinetically limits the nucleation of MOF crystals and creates textural mesopores among the aggregated polymorphic crystals[17]. The template directed strategy utilizes surfactants[19,21,22], block copolymers[23] or metal–organic assemblies[20,24] as sacrificial templates, which form nanodomains inside of MOF crystals and create large cavities once the templates are removed. These approaches provide facile synthetic routes towards hierarchical-pore MOFs. However, these methods typically introduce mesopores at the expense of MOF crystallinity, as the perturbation during MOF formation could result in partial amorphization[18] and the template removal step may cause collapse of the framework[15]. In addition, hierarchical-pore MOFs are usually obtained as crystalline powders, necessitating structural characterization by powder X-ray diffraction (PXRD) technique. In contrast, single crystalline MOFs allow for more precise structure characterization by single crystal X-ray diffraction (SCXRD), which provides direct structural evidence and therefore maximizes the understanding of structure–property correlation[25,26]. In this regard, coassembly of the ligand and its fragment into one MOF have brought new opportunities[27–29]. The linker fragment as a defect-generating ligand can be incorporated into MOF crystals during the synthesis, which introduces coordinatively unsaturated metal sites and functionalized mesopores. However, the concentration of defects introduced by this method is still limited because highly defective structures with relatively low stability cannot survive the harsh solvothermal synthetic conditions.

Herein, we report a method, namely linker labilization, to controllably increase the porosity and pore size of microporous MOFs. This strategy was inspired by the linker installation[30,31] and solvent-assisted ligand incorporation methods[32]. These two methods utilize MOFs with coordinatively unsaturated $Zr_6O_4(OH)_8(H_2O)_4$ clusters and postsynthetically incorporate linear linkers or terminal ligands on the $Zr_6$ clusters by replacing the terminal $–OH/H_2O$ with carboxylates (Fig. 1a). Linker labilization is a reverse process to linker installation: an MOF is initially constructed with coordinatively saturated clusters and a pro-labile linker; the pro-labile organic linkers are subsequently

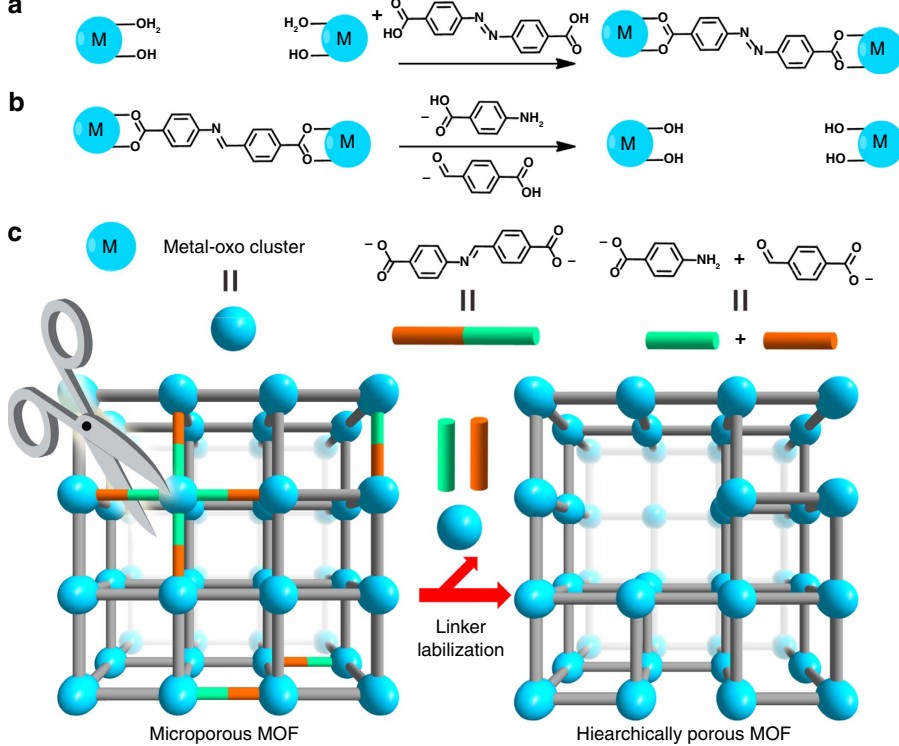

**Figure 1 | Schematic representations.** (**a**) Linker installation; (**b**) linker labilization; (**c**) hierarchically porous MOF developed by linker labilization.

labilized by splitting into two removable monocarboxylates under acidic conditions to introduce defects (Fig. 1b,c). Taking advantage of the acid stability of high valence metal-based MOFs and their strong tolerance towards defects[33–36], the crystallinity and porosity of hierarchical-pore MOFs are well maintained after acid treatment, water treatment and solvent removal. The maximum pore size determined by $N_2$ sorption analysis can be tuned from 1.5 to 18 nm by judicious control of the pro-labile-linker content during the MOF synthesis and the acid concentration during the linker labilization process. Linker labilization creates hierarchically porous structures with different levels of porosity, which allows for efficient diffusion of guests throughout the framework, making them promising materials for a wide range of applications including adsorption and catalysis.

## Results

**Labilizing Zr-MOFs by linker exchange.** Zr-MOFs were first selected as platforms for the implementation of the linker labilization strategy because of their superior stability and strong tolerance of missing-linker defects. An imine-based linker, denoted as CBAB (4-carboxybenzylidene-4-aminobenzate), was designed as the pro-labile linker, which can be easily dissociated into 4-amino benzoic acid and 4-formylbenzoic acid in a single step through hydrolysis. The equilibrium constant of CBAB hydrolysis reaction was measured to be 0.0601 in $N,N$-dimethylformamide (DMF), which increased to 4.36 in 0.5 M AcOH/DMF solution (AcOH = acetic acid). An azobenzene-based linker, namely AZDC (azobenzene-4,4'-dicarboxylate), was selected as a robust linker to support the framework. As AZDC and CBAB have very similar lengths and the same connectivity, they are expected to give rise to mixed linker MOFs when combined with $Zr_6$ clusters. The robust AZDC linkers can support the framework, whereas the CBAB can be removed by acid treatment to increase the MOF porosity. We attempted to synthesize mixed linker Zr-MOFs starting from a mixture of AZDC and CBAB in a one-pot reaction, but to no avail. Only a Zr–AZDC MOF was obtained, whereas the CBAB linker was not incorporated in the product. In fact, the synthesis of Zr-MOFs was carried out at a high temperature (120 °C) with an excess amount of carboxylic acid as a modulating reagent[37]. The CBAB presumably decomposed under such harsh conditions due to its acid sensitivity. To overcome this, postsynthetic linker exchange was adopted.

The Zr-AZDC-MOF, namely PCN-160, was synthesized as a template under solvothermal condition using trifluoroacetic acid as the modulating reagent. SCXRD revealed that PCN-160 crystallized in the cubic space group $Fm\bar{3}m$ (Supplementary Table 1). Each individual $Zr_6$ cluster was connected to 12 AZDC to form a UiO-type structure with face-centred cubic topology. It is noteworthy that Zr-AZDC MOFs isostructural to PCN-160 have been previously reported using different synthetic methods[38–41]. The linker exchange can be carried out under relatively low temperature without acid, which maintains the intactness of the imine bond in CBAB[42,43]. Considering their similarity, the AZDC in PCN-160 is expected to be substituted by CBAB through linker exchange.

Systematic studies were performed to investigate the linker exchange process. The linker exchange was carried out by incubating the crystals of PCN-160 in the CBAB/DMF solutions at 75 °C. As AZDC possesses a distinctive ultraviolet–visible (UV-vis) absorption peak at 432 nm, the linker exchange process can be easily monitored by UV-vis spectroscopy. The effects of incubation time and CBAB concentration in DMF solution were studied. As shown in Fig. 2a, the CBAB content in PCN-160 gradually increases as incubation time increases and levels off

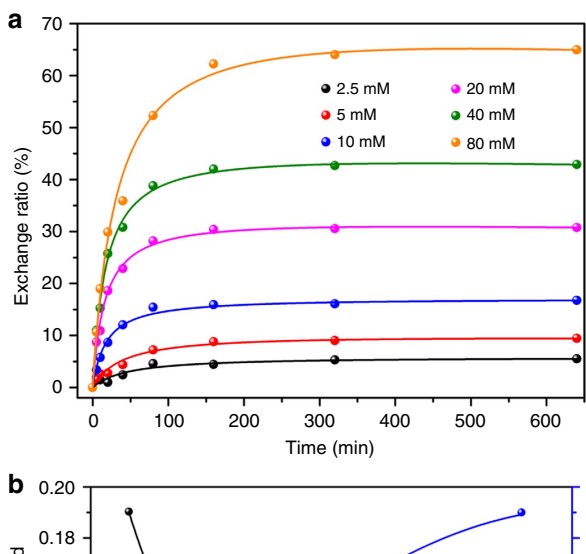

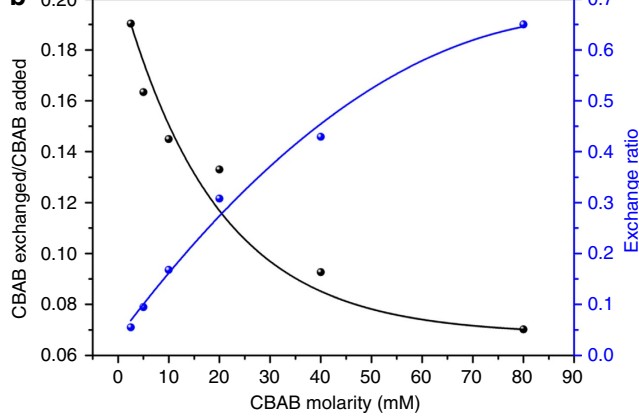

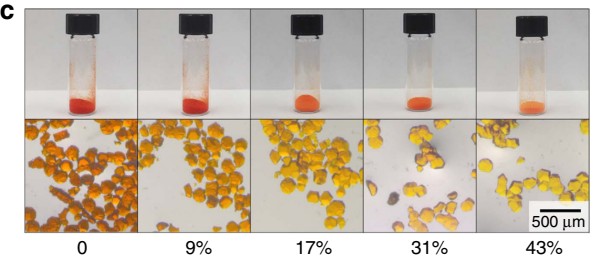

| 0 | 9% | 17% | 31% | 43% |

**Figure 2 | Linker exchange monitored by UV-vis spectra.** (**a**) AZDC concentration in supernatant as a function of incubation time in different concentrations of CBAB solutions; (**b**) relationship between CBAB exchanged/CBAB added, exchange ratio and CBAB concentration in solution; (**c**) images of the PCN-160 crystals with different exchange ratios.

after 150 min. Further elongation of incubation time does not increase the CBAB content, indicating that a dynamic equilibrium is reached between the solid and the solution. The exchange ratio, defined as moles of CBAB divided by the moles of organic linkers (CBAB + AZDC) in PCN-160, was calculated based on the AZDC concentration in the solution and the weight of the MOF material, which matches well with the [1]H-NMR result of digested samples. Upon completion of linker exchange, the MOFs with labile CBAB linkers are denoted as PCN-160-$R$%, where $R$% stands for the exchange ratio. The exchange ratios of PCN-160 vary from 5.5 to 65%, when incubated with CBAB solutions of different concentration. A positive correlation is observed between the CBAB concentration and the exchange ratio at equilibrium state. However, the ratio of CBAB exchanged to CBAB added decreases as the CBAB concentration increases, suggesting that stronger driving forces are needed to reach higher exchange ratios (Fig. 2b). The CBAB/DMF solution almost

saturates at 80 mM, which means the exchange ratio can reach as high as 65% without changing the stock solution. To drive the equilibrium of the system forward, the supernatant was exchanged with fresh saturated CBAB stock solution to remove the AZDC. Complete exchange can be realized by repeatedly exchanging the supernatant with fresh CBAB stock solution every 300 min for four times. Single crystal to single-crystal transformation is realized so that the structure of the resulting MOF after linker exchange can be precisely determined by SCXRD. The resulting MOF, PCN-160-100%, also crystallized in the cubic space group $Fm\bar{3}m$ with unit cell parameters of $a = b = c = 29.47$ Å, which is similar to that of PCN-160 ($a = b = c = 29.39$ Å). The UV-vis and $^1$H-NMR spectra of digested samples confirm that the AZDC is completely substituted by CBAB (Supplementary Table 2).

The supernatants after linker exchange were analysed by inductively coupled plasma mass spectrometry (ICP-MS), which showed no detectable $Zr^{4+}$. The weights of MOF materials were invariant before and after the linker exchange process, eliminating the possibility of dissolution or decomposition during the treatment. PCN-160-$R$% samples maintain single crystallinity as revealed by the PXRD patterns (Supplementary Fig. 2) and the microscope images of the respective single crystals (Fig. 2c). $N_2$ sorption isotherms also indicate that the porosity and structural integrity of PCN-160-$R$% are well-retained after the linker exchange (Fig. 3a). These results unambiguously show that the exchange ratio of PCN-160 can be precisely tuned with retention of framework integrity and porosity, which is essential for the linker labilization method.

**Creating hierarchical pores by linker labilization.** It is known that the CBAB linker easily dissociates into 4-amino benzoic acid and 4-formylbenzoic acid under acetic condition. The terminal benzoates are much more labile than the bridging linker, as they can be replaced by acetates or a pair of terminal –OH/$H_2O$

ligands[32]. Therefore, we proposed that the CBAB in PCN-160 could be partially removed by acetic acid treatment to create controllable missing-linker defects. Presumably the defect concentration can be affected by (a) exchange ratio of PCN-160 and (b) AcOH concentration. Therefore, control experiments were conducted to investigate the effects of each factor. PCN-160 samples with exchange ratios varying from 0 to 43% were prepared and treated with different concentrations of AcOH/DMF solution at 40 °C for 24 h (see Supplementary Methods for details). The resulting materials are denoted as PCN-160-$R$%-$C$, where $R$% stands for exchange ratio and $C$ represents the acid concentration. PXRD measurements showed that the MOF materials after acid etching possessed the same diffraction patterns as the parent PCN-160 (Supplementary Fig. 4). The microscope images of the respective single-crystals demonstrated that the crystal morphology was not affected by the linker labilization process (Supplementary Fig. 3). The porosity of each sample was estimated by $N_2$ sorption measurements. Before the gas sorption measurements, the MOF materials were washed thoroughly with DMF and water to remove any remaining acetic acid, and exchanged by acetone before thermal activation at 100 °C under vacuum. The $N_2$ sorption isotherms at 77 K demonstrated the formation of the hierarchically structured MOF materials with both micropores and mesopores (Fig. 3 and Supplementary Figs 5 and 6). In contrast to the type-I isotherm for the microporous PCN-160, type-IV isotherms were observed for PCN-160-$R$%-$C$ samples with hysteresis loops characteristic of large constricted mesopores.

The pore-size distribution of PCN-160-$R$%-$C$ samples were calculated from the $N_2$ adsorption isotherms using a density functional theory model. In general, the porosities and pore sizes of these MOFs show strong dependences on both exchange ratio and AcOH concentration. When 0.5 M acetic acid was used as the etching reagent, both porosities and pore sizes increased with the exchange ratios. For example, when the exchange ratio increases from 0 to 9%, an obvious mesopore with a diameter of 2.5 nm

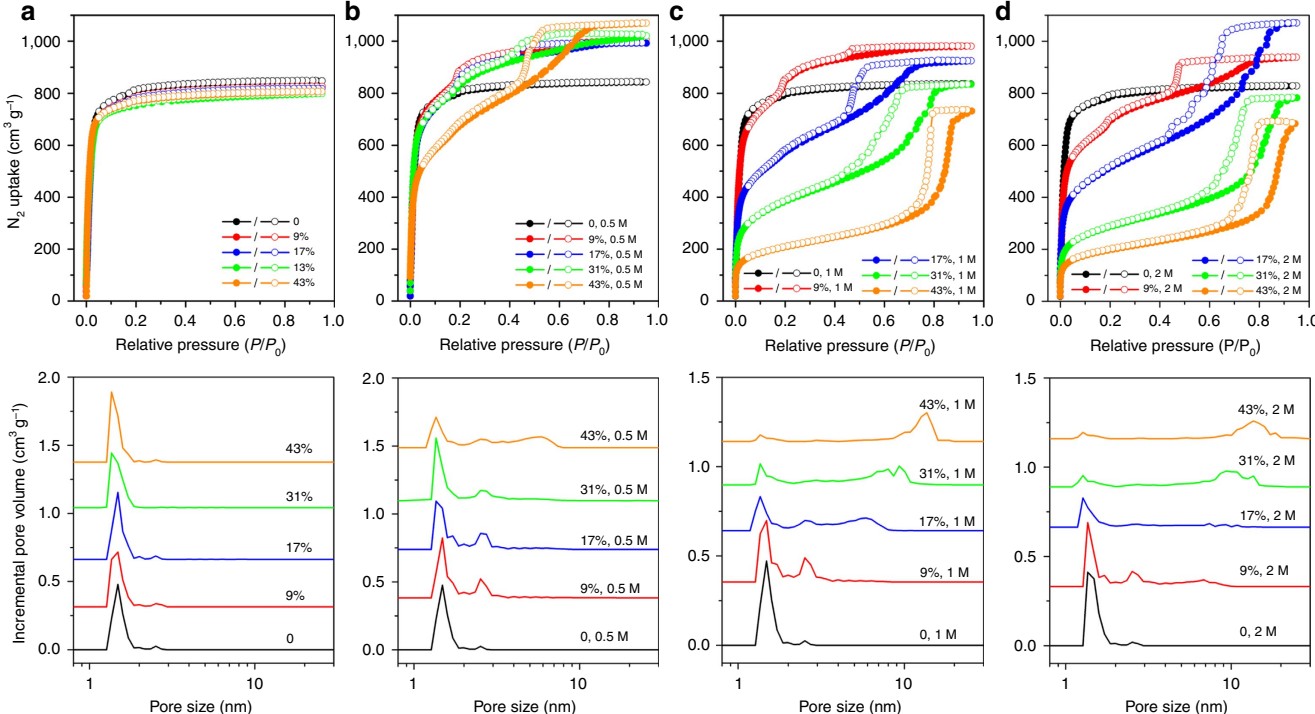

**Figure 3 | $N_2$ sorption isotherms and pore size distributions.** PCN-160-$R$%-$C$ treated with pure DMF (**a**), 0.5 M AcOH/DMF solution (**b**), 1 M AcOH/DMF solution (**c**) and 2 M AcOH/DMF solution (**d**).

emerges. Further increment of the exchange ratio creates a second mesopore as large as 5.4 nm. Although the pore size of the micropores (1.5 nm) is the same in all the hierarchical-pore MOFs, the pore volume of micropores decreases as the pore volume of mesopores increases, suggesting that the micropores are partially converted into mesopores through linker labilization. The MOF porosities, as indicated by the total $N_2$ uptake at 77 K and 1 bar, are dramatically enhanced by the linker labilization. This is attributed to the defects created by linker labilization, which increase the pore size while decreasing the material density.

The AcOH concentration also affects the porosity and pore size of hierarchical-pore MOFs. When PCN-160-31% was adopted as the starting material, the pore size of the resulting material was enlarged by increasing the AcOH concentration. The maximum pore size of PCN-160-31%-0.5 is only 2.5 nm, which is smaller than that of PCN-160-31%-1 (9.3 nm) and PCN-160-31%-2 (13.6 nm). However, the fraction of mesopores increases at the expense of a decrease in the total pore volume. The total $N_2$ uptake of PCN-160-31%-2 (782 $cm^3 g^{-1}$) decreased by 27% compared with that of PCN-160-31%-0.5 (1,066 $cm^3 g^{-1}$). Meanwhile, a slightly broader PXRD peak and increased PXRD background were observed for PCN-160-31%-2 (Supplementary Fig. 4d). These results indicate that high AcOH concentration and exchange ratio create a large concentration of defects, which forms large mesopores and partially destroys the framework integrity at the same time. Therefore, the porosity of PCN-160-$R$%-$C$ represents a delicate balance between defect concentration and framework stability. To maintain the framework integrity, the AcOH concentration can reach as high as 2 M, whereas the exchange ratio is no more than 43%. A contour plot was generated to describe the relationship among AcOH concentration, exchange ratio and the maximum pore size (Fig. 4). In general, the pore size is positively related to the AcOH concentration and exchange ratio (Supplementary Table 3). Therefore, the pore size of PCN-160 can be tuned by judicious control of exchange ratio and AcOH concentration, providing a versatile platform for pore size-directed applications.

**Defect analysis**. To investigate the origin of mesopores created by linker labilization, a series of experiments were performed in combination with molecular simulations. First, we seek to determine the composition change of PCN-160 before and after linker labilization. The PCN-160-31%, PCN-160-31%-0.5, PCN-160-31%-1 and PCN-160-31%-2 samples were precisely weighted, dissolved in $K_2CO_3$ solution and analysed by UV-vis spectra, to determine the weight per cent of AZDC and CBAB in MOF materials (see Supplementary Methods for details). As shown in Supplementary Fig. 7, the weight per cent of CBAB in PCN-160-31% decreases after acid treatment, indicating that the labile CBAB linkers are partially removed by acid etching. As a control experiment, PCN-160 was treated with AcOH under identical condition, which did not show obvious weight loss. The connection numbers of the Zr clusters in the MOF samples were calculated according to the weight percent of each linker (Supplementary Table 4). The $Zr_6$ clusters in PCN-160 and PCN-160-31% are ∼12-connected, which are reduced to ∼9-connected after acid treatment. The concentrations of CBAB and AZDC in the solution after the linker labilization process were also analysed by UV-vis spectra. The concentration of CBAB is much higher than that of AZDC, indicating the removal of labile CBAB linker under acidic conditions (Supplementary Fig. 8).

SCXRD is a powerful and ubiquitous technique to definitively elucidate structures at the molecular level. Given that PCN-160-$R$%-$C$ could undergo linker exchange and linker labilization with retention of single crystallinity, direct evidence of the structural change can be gathered by SCXRD. Although the exact identity of defect sites is unknown, the missing linkers would most likely be replaced by a pair of –OH/$H_2O$ groups to compensate for the charge loss from the linkers. In a $Zr_6$ cluster terminated by –OH/$H_2O$, each terminal –OH/$H_2O$ ligand is separated as far as possible to reduce the steric hindrance and Coulomb repulsion. Hence, the O–Zr–O angle is ∼100° (refs 30,31,44). However, restricted by the O–O distance of carboxylates, the O–Zr–O bond angle in a 12-connected $Zr_6$ cluster is about 120°. Therefore, the obvious distinction of O from carboxylates and –OH/$H_2O$ ligands by SCXRD allows for the direct 'observation' of defects in the resolved single-crystal structure. A comparison of PCN-160-31% and PCN-160-31%-2 single-crystal structure clearly indicates the emergence of defects after AcOH etching. The terminal –OH/$H_2O$ ligands in PCN-160-31% are almost invisible by crystallography. However, after AcOH treatment, the terminal –OH/$H_2O$ ligands can be unambiguously differentiated from carboxylates (Supplementary Fig. 1). The occupancies of the carboxylate linkers were refined as 75% and the terminal –OH/$H_2O$ was refined with occupancies of 25%, in agreement with the connection number calculated from UV-vis data.

Further evidence consistent with creation of defects is provided by the diffuse reflectance infrared Fourier transform spectroscopy measurements (Supplementary Fig. 9a). According to the literature, the sharp peak appearing at 3,666 $cm^{-1}$ is assigned to the terminal –OH groups and bridging $\mu_3$-OH groups, and the broad band centred at 3,446 $cm^{-1}$ corresponds to the $H_2O$, which either binds to $Zr^{4+}$ as a terminal ligand or interacts with the $Zr_6$ clusters through hydrogen bonds[32,45]. In the nearly defect-free PCN-160-31%, a sharp peak appeared at 3,666 $cm^{-1}$ and a relatively weak peak was observed at 3,446 $cm^{-1}$, corresponding to the $\mu_3$-OH groups and lattice $H_2O$, respectively. After acid treatment, the –OH peak at 3,666 $cm^{-1}$ becomes broader with a shoulder appearing at 3,698 $cm^{-1}$, which is consistent with terminal -OH groups observed for NU-1000 (ref. 32). Meanwhile, the $H_2O$ peak becomes intense and broad, due to the presence of terminal –$H_2O$ on the $Zr_6$ cluster, which in turn attracts peripheral $H_2O$ molecules through hydrogen bonding. Thermogravimetric analysis (TGA) data further confirms the creation of defects by replacing linkers with

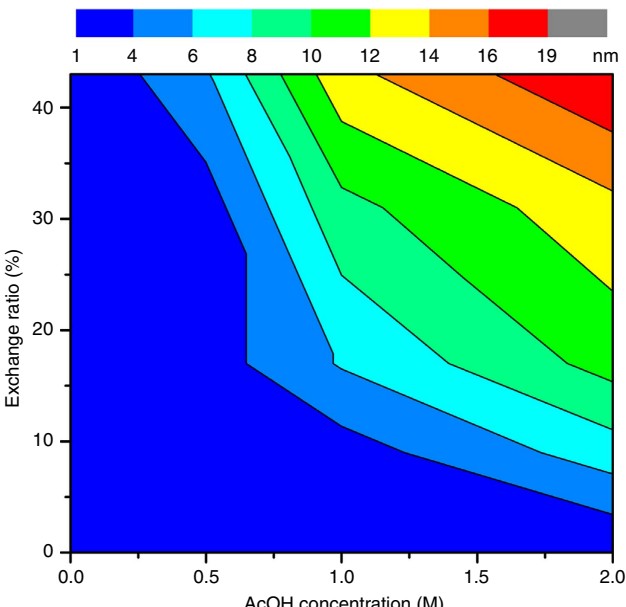

**Figure 4 | Control of pore size.** Maximum pore size of PCN-160-$R$%-$C$ as a function of exchange ratio and AcOH concentration.

terminal –OH/$H_2O$ (Supplementary Fig. 9b). For PCN-160-31%, no obvious weight loss was observed before 300 °C, indicating a nearly defect-free structure. The acid etched samples showed a weight loss of 8% before 300 °C, which is attributed to the removal of the coordinated $H_2O$ on the cluster and the hydrogen bonded water species in the cavity. In addition, the weight loss corresponding to the thermal decomposition of organics became smaller after acid treatment, suggesting that organic linkers were partially removed.

**Proposed formation mechanism of hierarchical pores.** With the material composition in hand, we further turned our efforts to investigate the origin of the hierarchical pores by molecular simulation. Intuitively, the CBAB and AZDC linkers are expected to be randomly distributed in PCN-160, considering the similar length and connection of two linkers. The acid treatment partially removes CBAB, leaving random missing-linker defects. A missing-linker model was built based on this assumption, in which a stoichiometric $3 \times 3 \times 3$ supercell of the face-centred cubic PCN-160 was taken and randomly selected linkers were removed (see Supplementary Methods for details). Pore size distribution was estimated by Zeo++ with $N_2$ as a probe (radius 1.82 Å). The maximum pore size of PCN-160 was simulated to be 1.5 nm, which matched well with the pore diameter from $N_2$ sorption isotherms (1.5 nm). According to the composition determined by UV-vis and $^1$H-NMR techniques, about 25% of linkers in the PCN-160-31%-0.5 were removed by acid etching. However, the mesopores were not observed in the simulated pore size distribution even though 25% of linkers were removed from the model. These results contradicted the missing-linker model and suggested the possible existence of missing-

cluster defects in PCN-160-*R*%-*C*. In fact, the missing-cluster defects have been observed and thoroughly studied in UiO-66 (ref. 46). The missing of $Zr_6$ clusters along with the linkers around them locally created an eight-connected net with $ReO_3$ (reo) topology. The model with reo defects was built by removing a suitable number of randomly selected $Zr_6$ clusters and linkers. The simulated maximum pore size (2.7 nm) corresponds to the experimentally observed 2.5 nm mesopore (Supplementary Figs 10 and 11). Further evidence is provided by the ICP-MS data of the supernatant after acid treatment, which indicates the existence of $Zr^{4+}$ as a result of cluster removal (Supplementary Fig. 8). Therefore, the 2.5 nm mesopore can be explained by the reo defect, in which a $Zr_6$ cluster is removed along with linkers around it.

Furthermore, multiple adsorption/desorption steps were observed for the PCN-160-31%-1, indicating that different kinds of textural mesopores coexist in the hierarchical-pore MOFs. Although the 2.5 nm mesopore can be explained by the reo defects, the second mesopore (>5 nm) is difficult to explain on the molecular level. Scanning electron microscope (SEM) and transmission electron microscope (TEM) were adopted to reveal the morphology of the hierarchical-pore materials, to uncover the formation mechanism of the large mesopores (Supplementary Figs 12–15). As expected, the PCN-160-31% crystals before acid treatment show a very smooth surface in the SEM images. SEM and TEM images of PCN-160-31%-0.5 show the appearance of mesopores as indicated by riddled edges. Different from PCN-160-31% and PCN-160-31%-0.5, PCN-160-31%-1 shows a sponge-like morphology, corresponding to the 5.4 nm mesopores from $N_2$ sorption isotherms. In the case of PCN-160-31%-2, a pomegranate-type structure is observed, in which the

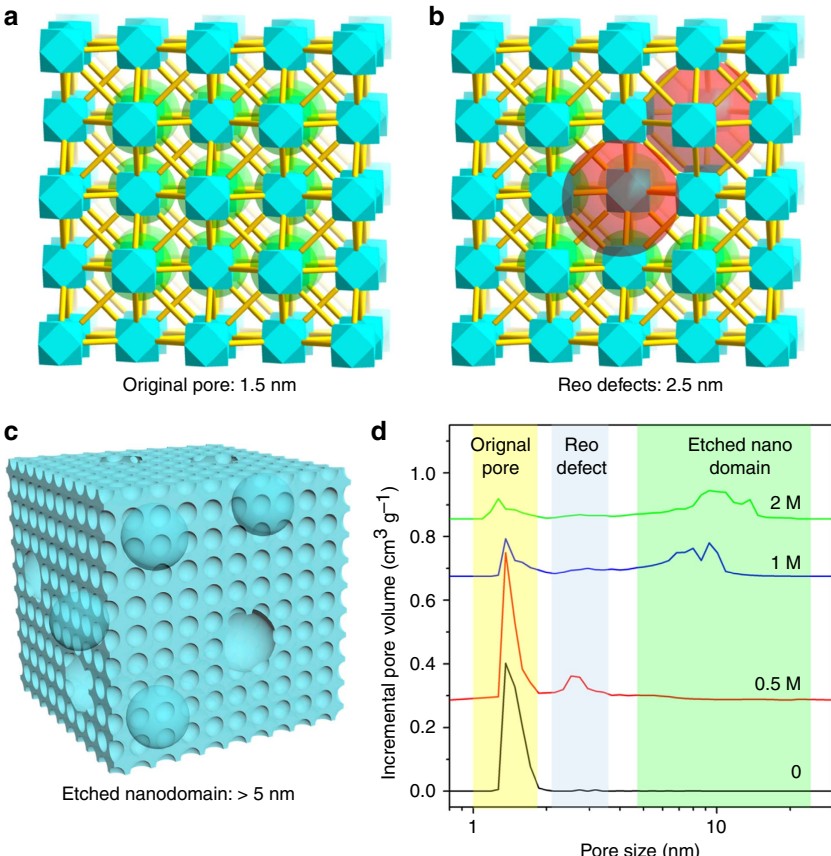

**Figure 5 | Schematic illustration of formation mechanism.** Formation mechanism of the micropores (**a**), small mesopores (**b**) and large mesopores (**c**); (**d**) pores size distribution of PCN-160-34% treated with different amounts of acid, highlighting the origin of different pores.

microcrystalline MOF forms nanodomains. On the basis of the above results and literature precedents[17,18,20,46], a possible mechanism for the formation of large mesopores ($> 5$ nm) is proposed. The linker labilization initially induces missing-linker defects by partially removal of labile CBAB linkers, which reduced the connection number of $Zr_6$ clusters. As a result, some $Zr_6$ clusters and the linkers around are removed to form missing-cluster defects (reo defects), corresponding to 2.5 nm mesopores (Fig. 5a,b). As the defect sites are relatively labile compared with defect-free structures, further acid etching gradually dissolves the materials from the defect sites as weak points, which enlarges the mesopores while maintaining the crystallinity (Fig. 5c). This explains the observation that the small mesopore volume decreases as the large mesopore emerges. A sponge-like structure is therefore obtained by enlarging the 2.5 nm mesopores. As etching of the crystalline framework proceeds, the framework becomes fragmented, resulting in the pomegranate-type of morphology. In other words, the linker labilization process partially dissolves the MOF crystals from the inside out, creating large mesopores inside the particle while leaving the particle size visually intact.

Based on this mechanism, the effects of exchange ratios and acid concentrations on the pore architecture of PCN-160 can be rationalized. The exchange ratio controls the concentration of the weak points in MOF crystals where the crystal dissolution initiates. Meanwhile, the acid concentration affects the rate of dissolving on each weak point. With high exchange ratio and low acid concentration, the concentration of weak points is high, whereas the dissolving rate is slow so that the formation of reo defects are favoured. This corresponds to the observation that the mesopores with diameter of 2.5 nm were observed at low acid concentration. On the other hand, the large mesopores ($>5$ nm) are usually created by the extension of small mesopores (2.5 nm) at high acid concentration. It should be noted that defects can be introduced and controlled by varying the modulating acid during the MOF synthesis[46]. However, the defect concentration and mesopore volume introduced in a one-pot reaction are still limited, because the relatively low stability of highly defective structures which cannot survive the harsh solvothermal reaction conditions. In contrast, linker labilization is carried out at low temperature under mild conditions, which maintained the porosity and crystallinity of MOFs to the maximum extent. Therefore, the pore sizes and pore size distributions of PCN-160 can be precisely controlled by adjusting the exchange ratios and acid concentrations, providing a versatile platform for a variety of potential applications, especially when specific pore architectures are needed.

**Enhancing guest adsorption and catalytic performance**. To check the accessibility of the created pore spaces by linker labilization, a guest uptake study was performed. PCN-160-17% (1.5 nm), PCN-160-17%-0.5 (2.5 nm) and PCN-160-17%-1 (5.8 nm) with distinct maximum pore size were selected as representative examples of the hierarchical-pore MOFs. A series of guest molecules with different sizes, including $[Cr_2O_7]^{2-}$ ($0.3 \times 0.5$ nm), $[Ni_4(H_2O)_2(PW_9O_{34})_2]^{10-}$ (polyoxometalate or POM, $1.1 \times 1.4$ nm) and $Cu_{24}(BDC-OH)_{24}(H_2O)_{24}$ (metal-organic polyhedron or MOP, 3.1 nm), were selected as the probe molecules. The molecular dimension of $[Cr_2O_7]^{2-}$ is much smaller than the window size of PCN-160-17%. However, the $[Cr_2O_7]^{2-}$ uptake is not only dependent on the pore volume, but also strongly affected by the defect concentration, because the missing-linker sites in Zr-MOFs are reported to bind strongly with anions, such as arsenate, selenate and dichromate (Supplementary Fig. 16)[47–49]. As shown in Supplementary

Fig. 17a, the PCN-160-17%-0.5 and PCN-160-17%-1 with much higher concentration of missing-linker defects show higher dichromate uptake compared with that of PCN-160-17%, demonstrating the effect of defects. Although the micropore size (1.5 nm) in PCN-160-17% is large enough for POMs, the diffusion of POMs in the MOF cavity is limited by the small window size (0.9 nm), which is smaller than the molecular dimensions of the POM ($1.1 \times 1.4$ nm). In contrast, the hierarchical pore architecture of PCN-160-17%-0.5 and PCN-160-17%-1 allows quick diffusion of large molecules through mesopores, which finally reach the storage sites within micropores. Therefore, the POM uptake and rate of adsorption are enhanced by linker labilization (Supplementary Fig. 17b). To recognize different mesopore sizes of PCN-160-17%-0.5 and PCN-160-17%-1, the adsorption of the MOP was tested. It was found that the PCN-160-17%-1 with 9.3 nm mesopores have a much higher MOP uptake than PCN-160-17%-0.5 and PCN-160-17%, indicating the contribution of large mesopores (Supplementary Fig. 17c).

The hierarchical porous architectures created by linker labilization allow for efficient diffusion and adsorption of large molecules, making them promising materials for a wide range of applications, including adsorption, separation and catalysis. To investigate the effect of linker labilization on the catalytic performance of MOFs, we chose CYCU-3 as a platform[50]. CYCU-3 is a robust MOF constructed from 1D Al–O chains connected by linear AZDC linkers. It features a hierarchical micro-mesopore architecture with hexagonal channels (3.0 nm in diameter) and triangular channels (1.5 nm in diameter). The large hexagonal channels are suitable for enzyme encapsulation and the triangular channels allow for efficient substrate diffusion[51–53]. However, the walls of the channels in CYCU-3 are too condensed to allow any substrate diffusion between neighbouring channels. Therefore, the accessibility of the immobilized enzymes are largely suppressed[54,55]. We propose that linker labilization can create defects on the walls of the channels in CYCU-3, which act as widows that connects the channels throughout the crystal to facilities the diffusion of reactants and products.

Linker labilization was carried out on CYCU-3 by replacing 22% of the AZDC linkers with labile CBAB linkers and subsequently removing the labile linkers by treating with 0.2 mM HCl in DMF to create a defected MOF (CYCU-3D). The crystallinity of CYCU-3D samples is well maintained after linker labilization as revealed by the PXRD patterns (Supplementary Fig. 18). $N_2$ sorption isotherms of CYCU-3D indicate the growth of mesopores with a diameter of $\sim 7$ nm along with the decrease of small mesopore volume (3 nm in diameter, Supplementary Fig. 19). This is consistent with the observation in PCN-160 system that the large mesopores were created by the extension of small mesopores through acid etching. Cytochrome $c$ (Cyt $c$) with a molecular dimension of $2.6 \times 3.2 \times 3.3$ nm were immobilized in CYCU-3 and CYCU-3D by treating the activated MOF crystals with aqueous solution of Cyt $c$ (300 μM) at 25 °C. The uptake of Cyt c was determined by UV-vis spectroscopy, which shows a maximum loading of 23.3 mmol g$^{-1}$ for CYCU-3 and 14.5 mmol g$^{-1}$ for CYCU-3D after 48 h. Pore-size distributions CYCU-3 and CYCU-3D after enzyme loading suggests that the enzyme occupies the hexagonal channels, leaving the triangular channels poised for substrates (Supplementary Fig. 20).

Before catalytic reaction, the solid samples were immersed in water for 2 days during which the solvent was decanted and freshly replenished five times, to ensure full removal of any loosely bound Cyt $c$. UV-vis spectroscopy reveal that no detectable enzyme leaching in water within 2 days after washing. To evaluate the accessibility of Cyt $c$ after

immobilization in different MOFs, the oxidation of *o*-phenylenediamine and 2,2′-azino-bis(3-ethylbenzothiazoline-6-sulphonic acid) (ABTS) catalysed by Cyt *c* were performed (see Supplementary Methods for details). The molecular size of *o*-phenylenediamine ($0.5 \times 0.5$ nm) is relatively small so that it could diffuse in the hexagonal channels to reach the active sites of encapsulated enzymes. Therefore, the Cyt *c* immobilized in CYCU-3 and CYCU-3D both show similar $k_{cat}$, $K_m$ and $V_{max}$ values to the free enzyme, indicating a comparable catalytic performance (Supplementary Fig. 21). However, the catalytic activity of Cyt *c* immobilized in CYCU-3 dramatically decreased for bulky substrates (Supplementary Fig. 22). As shown in Fig. 6, the relative activity for ABTS oxidation is reduced by 90% when Cyt *c* was encapsulated in CYCU-3, which may be caused by the diffusion control. Considering the sizes of ABTS ($0.7 \times 1.6$ nm) and Cyt *c*, the hexagonal channels are expected to be blocked by the encapsulated Cyt *c*, leaving no space for the diffusion of ABTS. The ABTS cannot access the enzyme from the triangular channels, as the walls of the channels are too condensed. Therefore, the diffusion of ABTS in CYCU-3 are largely restricted, which in turn results in the low activity of the encapsulated Cyt *c*. However, the activity of Cyt *c* in CYCU-3 increased almost five times after linker labilization. The Cyt *c* immobilized in CYCU-3D shows dramatically enhanced $k_{cat}$ and $V_{max}$ compared with the ones in CYCU-3, highlighting the beneficial effect of hierarchical porous structure on the catalytic performance (Supplementary Table 5). The defects created by linker labilization not only creates large mesopores (7 nm in

diameter) but also forms windows on the wall of the channels, allowing for efficient substrate diffusion between neighbouring channels. Therefore, the substrates can reach the active centre of Cyt C through the triangular channels even though the hexagonal channels are not accessible. This result corroborates well with the findings by Hupp and colleagues[54] in which they demonstrated the critical role of the hierarchical structure of NU-1000 for enzyme encapsulation applications. The hierarchical pore architecture created by linker labilization represents an integration of micropores, mesopores and windows that connect them throughout the crystal, which allows binding of the enzyme and diffusion of substrates.

In conclusion, a linker labilization method was developed to increase the porosity and pore size of microporous MOFs by creating crystal defects controllably. MOFs were constructed with various amount of pro-labile linkers, which were partially removed by acid etching to create mesopores with tunable sizes. The maximum pore size by $N_2$ sorption measurements is tuned from 1.5 to 18 nm by judicious control of pro-labile-linker content and acid concentration. Experiments in combination with molecular simulations revealed the formation mechanism of the mesopores. The beneficial effect of hierarchical porous structure on the adsorption properties and catalytic performances was demonstrated. We believe linker labilization may provide a feasible and versatile method to enlarge MOF pore size, which promises potential applications in guest adsorption/separations, heterogeneous catalysis, drug delivery, and sensing.

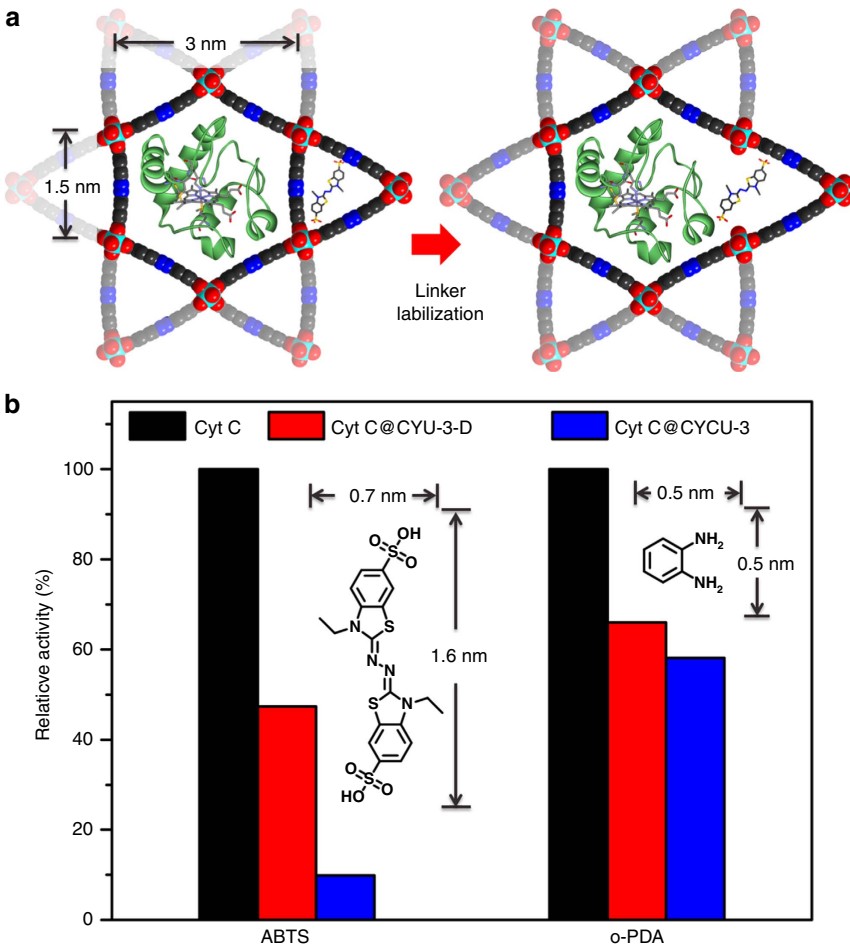

**Figure 6 | Enhancing activity of immobilized enzyme.** (**a**) Illustration of the accessibility of enzymes in CYCU-3 and CYCU-3D; (**b**) relative activity of Cyt *c*, Cyt *c* @CYCU-3 and Cyt *c* @CYCU-3D for the oxidation of ABTS and *o*-phenylenediamine (o-PDA).

## Methods

**Synthesis of PCN-160.** ZrCl$_4$ (200 mg), H$_2$AZDC (100 mg), trifluoroacetic acid (1.0 ml) and DMF (20 ml) were charged in a Pyrex vial. The mixture was heated in 120 °C oven for 72 h. After cooling to room temperature, the red crystals were harvested (95 mg, yield: 67%).

**Synthesis of PCN-160-$R$%.** PCN-160-$R$% ($R = 6$, 9, 17, 31, 43 and 65) were synthesized by linker exchange of PCN-160 with CBAB solution of different concentrations (2.5, 5, 10, 20, 40 and 80 mM, respectively). In general, crystals of PCN-160 (100 mg) were incubated with the solution of H$_2$CBAB in DMF (20 ml) at 75 °C for 10 h. PCN-160-100% were synthesized by repeatedly exchange the supernatant with fresh CBAB stock solution (80 mM) every 5 h for four times. The crystals of PCN-160-$R$% were collected by filtration and washed with fresh DMF for three times.

**Synthesis of PCN-160-$R$%-$C$.** PCN-160-$R$%-$C$ ($C = 0.5$, 1 and 2) was synthesized by treating the crystals of PCN-160-$R$% with different concentrations of AcOH in DMF solution (0.5, 1 and 2 mM, respectively). In general, crystals of PCN-160-$R$% (100 mg) were incubated with the solution of AcOH in DMF (20 ml) at 40 °C for 24 h. The crystals were collected by filtration and washed with fresh DMF and water to remove any remaining acetic acid.

**Characterization.** Gas sorption measurements were conducted using a Micrometrics ASAP 2020 system. PXRD was carried out with a Bruker D8-Focus Bragg–Brentano X-ray Powder Diffractometer equipped with a Cu sealed tube ($\lambda = 1.54178$ Å) at 40 kV and 40 mA. SCXRD was measured on a Bruker Venture CMOS diffractometer equipped with a Cu-$K_\alpha$ sealed-tube X-ray source ($\lambda = 1.5406$ Å). NMR data were collected on a Mercury 300 spectrometer. Ultraviolet–visible absorption spectra were recorded on a Shimadzu UV-2450 spectrophotometer. ICP-MS data were collected with a Perkin Elmer NexION 300D ICP-MS. TGA was conducted on a TGA-50 (Shimadzu) thermogravimetric analyser. Infrared measurements were performed on a Shimadzu IR Affinity-1 spectrometer. TEM experiments were conducted on a FEI Tecnai G2 F20 ST microscope (America) operated at 200 kV. Field-emission SEM images were collected on the FEI Quanta 600 field-emission SEM (America) at 20 KV.

**Data availability.** The X-ray crystallographic coordinates for structures reported in this study have been deposited at the Cambridge Crystallographic Data Centre (CCDC), under deposition number CCDC 1525246–1525249. These data can be obtained free of charge from the CCDC via www.ccdc.cam.ac.uk/data_request/cif. All relevant data supporting the findings of this study are available from the corresponding authors on request.

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

## Acknowledgements

The gas adsorption–desorption studies of this research was supported by the Center for Gas Separations Relevant to Clean Energy Technologies, an Energy Frontier Research Center funded by the US Department of Energy, Office of Science, Office of Basic Energy Sciences under Award Number DE-SC0001015. Structural analyses were supported as part of the Hydrogen and Fuel Cell Program under Award Number DE-EE-0007049. This computational work was funded by the Robert A. Welch Foundation through a Welch Endowed Chair to HJZ (A-0030). Texas A&M Supercomputing Facility was acknowledged to provide computing resources. The Distinguished Scientist Fellowship Program (DSFP) at KSU is gratefully acknowledged for supporting this work. We also acknowledge the financial supports from U.S. Department of Energy Office of Fossil Energy, National Energy Technology Laboratory (DE-FE0026472) and Qatar National Research Fund (NPRP9-377-1-080). Professor Di Sun from Shandong University is acknowledged for his assistance in crystallography. S.Y. also acknowledges the Texas A&M Energy Institute Graduate Fellowship Funded by ConocoPhillips and Dow Chemical Graduate Fellowship.

## Author contributions

Original idea was conceived by H.-C.Z. and S.Y.; experiments and data analysis were performed by S.Y., L.Z., J.-S.Q., J.L., L.F. and X.W.; structure characterization was performed by J.-S.Q.; molecular simulations were performed by L.H. and T.C.; manuscript was drafted by H.-C.Z., S.Y., J.-S.Q., L.F., M.B. and A.A. All authors have given approval to the manuscript.

## Additional information

**Competing interests:** The authors declare no competing financial interests.

**Publisher's note**: 

