## [Peer Review File · Nature Communications]

Reviewers' comments:

Reviewer #1 (Remarks to the Author):

This manuscript from the Zhou group describes a post-synthetic modification (PSM) approach to the generation of mesoporous materials that is quite ingenious and likely to be generally relevant. Specifically, an as prepared porous material, PCN-160, is ligand substituted with a linker ligand that is subsequently subjected to hydrolysis so that it can no longer serve as a linker. In addition, the authors demonstrate ability to conduct a further round of PSM that leads to incorporation of other moieties including catalytically active species. I am therefore supportive of acceptance but there are some issues to be resolved:

- Most importantly, the suggestion from the figures (i.e. Scheme 1, Figure 4) is that the substitution of ligands in the first round of PSM is non-random and that this results in predictable pore size and cavities. However, I find this hard to accept without supporting evidence. At this point there is no question that larger pores are created but they could be of broader size distribution than implied if the ligand substitution process is random (i.e. the extra pore size is due more to linker loss than cluster loss). The SEM images presented in SI are not resolved to a level of detail that is good enough to look at the surface properly.
- Related to the above point, there could be a higher distribution of PSM at the surface if linker substitution occurs from the outside in – all we know at this point is the average pore size distribution.
- Scheme 1 is a bit misleading since it implies that terephthalate is used as a ligand when this is not the case.
- It is good that the authors are not focused just upon pore size but also upon what can be done with larger pore size.
- There are an awfully large number of self-citations. The post-doc who probably wrote the paper should do a better job at citing earlier literature.

To summarize, the science behind this work is novel and of broad interest but not yet fully substantiated. In my opinion this manuscript would be suitable for publication in Nature Communications if experiments were to be conducted that address the randomness (or lack of randomness) of the mesopores.

Reviewer #2 (Remarks to the Author):

This manuscript reports a novel and exciting strategy to construct hierarchically porous MOFs by use of ligand labilization. The strategy has well demonstrated and systematically studied on the platform of Zr-based PCN-160. As shown in the study, the chemical instability of imine-derived ligand provides a great opportunity to generate crystal defects or hierarchical pores. Additionally, the linker labilization proves to be an effective and efficient way to enhance guest adsorption capacity and catalytic performances. By use of MOF instability and thermodynamics, this study will generate a new perspective for MOF community and stimulate more interest toward MOFs and other materials. It warrants publication in Nature Communications and there are some minor comments as listed below.

1. The quality of TEM images are not quite good; if possible, the authors should collect some high-resolution TEM images to present more clearly direct evidence for the presence of mesopores in the MOFs. As hinted from the SEM images, the presence of pores larger than 10 nm in some MOFs herein is likely due to the voids of some decomposed MOF particles similar to mesoMOF-1 reported from this research group, which however is not cited (J. Am. Chem. Soc., 2006, 128, 16474-16475). The authors should be cautious about the statement "The maximum pore size can be tuned from 1.5 nm to 18 nm".
2. The unit for "Maximum pore size" in the caption of Table S3 should be "angstrom" but NOT

“nm”.

3. To better quantify the amount of defects in the MOF, the authors could conduct the titration experiments by referring to some works published by Hupp and Wei Zhou.

4. The literature citation needs certain improvement. Some references are not quite related, e.g. Ref. 8, 9, 13, 30. Some citations are similar, e.g. Ref. 3&4, Ref. 5&6, and Ref. 37,38,&39 (the year in Ref. 4 and Ref. 6 should be 2012). Ref. 58 should be *J. Am. Chem. Soc.*, 2016, 138, 8052–8055. Some other related literatures are missing: e.g. some recent papers regarding the pore size expansion in MOFs (*J. Am. Chem. Soc.* 2016, 138, 12045-12048; *J. Am. Chem. Soc.* 2015, 137, 10508-10511); one more paper regarding the synthesis of Zr-AZDC MOF (*Sci. China Chem.*, 2016, 59, 980-983); the first work advocating the benefits of employing hierarchically porous MOFs for enzyme immobilization (*J. Am. Chem. Soc.*, 2011, 133, 10382-10385); the first work about the size-selective biocatalysis utilizing hierarchically porous MOF (*Inorg. Chem.*, 2012, 51, 9156-9158); some review articles regarding the enzyme immobilization in MOFs (*Mater. Horiz.*, 2017, DOI: 10.1039/C6MH00312E; *Dalton Trans.*, 2016, 45, 9744-9753).

Reviewer #1:

1. Most importantly, the suggestion from the figures (i.e. Scheme 1, Figure 4) is that the substitution of ligands in the first round of PSM is non-random and that this results in predictable pore size and cavities. However, I find this hard to accept without supporting evidence. At this point there is no question that larger pores are created but they could be of broader size distribution than implied if the ligand substitution process is random (i.e. the extra pore size is due more to linker loss than cluster loss). The SEM images presented in SI are not resolved to a level of detail that is good enough to look at the surface properly.

Response: Thank you for the valuable suggestions. We believe that CBAB and AZDC linkers are randomly distributed in PCN-160-R% after linker exchange. This is supported by the following evidences. Firstly, the microscopic images of PCN-160-R% crystals show a uniform color throughout the crystal, eliminating the existence of macroscopic domains of each linker (Fig. 1c and Supplementary Fig. S3). Since AZDC with azobenzene moiety has distinguishable red color whereas the color of CBAB is pale yellow, macroscopic domains with different links ratios should have distinct colors. The crystals were further dissected into pieces, which also possess a uniform color (Figure below). Secondly, the uniform distribution of linkers can be determined by UV-vis. As an example, the crystals of PCN-160-31% were dissected into unequal parts. The smaller parts and the larger parts were collected respectively, and digested for UV-vis spectra analysis. The linker ratio determined by UV-vis spectra are almost the same for each part (33% for larger parts and 30% for smaller parts). If there is a higher exchange ratio on the surface, a different link ratio would be expected.

Acid treatment randomly removed the labile CBAB linkers, creating missing linker defects. However, the missing linker defects do not create noticeable change in pore size distributions, which is proved by the molecular modeling and simulations (Supplementary Fig. 11). When the connection number of a cluster is reduced to a certain extent, it is likely that the whole cluster including the linkers around it can be removed by acid etching, creating missing cluster defects (**reo** defects). Indeed, a small amount of AZDC linker was observed in the supernatant after linker labilization process as proved by UV-vis spectra (Supplementary Fig. 8). This suggested that a Zr-cluster and AZDC linkers around it were removed to create **reo** defects. Zr-MOFs with **reo** defects have been previously reported and they are believed to be metastable structures (*Nat. Commun.* **2014**, 8, 4176). Therefore, a relatively uniform pore structure and narrow pore size distribution was observed for the 2.5 nm pore (corresponding to **reo** defects). The large mesopores indeed have a broader size

distribution (from 5.4 to 10.8 nm for PCN-160-31%-1), which is in accordance with our assumption that the ligand substitution process is random. Limited by the resolution of SEM spectroscopy, the SEM images only show riddled edges of particles after acid etching. The TEM images clearly shows the sponge-like morphology formed by linker labilization.

Fig. 4 is an illustration of missing cluster defects whereas the missing linker defects is omitted for clarity. Scheme 1 is a bit misleading since the missing linker defects are not shown. It has been modified in the revised manuscript. The following explanations have been added in the revised manuscript.

“The linker labilization initially induces missing-linker defects by partially removal of labile CBAB linkers, which reduced the connection number of Zr_6 clusters. As a result, some Zr_6 clusters and the linkers around are removed to form missing-cluster defects (**reo** defects), corresponding to 2.5 nm mesopores (Fig. 4a and b).”

2. Related to the above point, there could be a higher distribution of PSM at the surface if linker substitution occurs from the outside in – all we know at this point is the average pore size distribution.

Response: The following experiments have been carried out to show the uniform distribution of linkers after linker exchange. Firstly, the microscopic images of PCN-160-R% crystals during linker exchange were taken (Figure below). After 20 minutes, the surface of the crystals are lighter in color, indicating that the exchange occurs from outside in. An equilibrium is reached within 300 min when a uniform color is observed throughout the crystal. Further elongation of incubation time does not change the color of the crystals. Since AZDC with azobenzene moiety has distinguishable red color whereas the color of CBAB is pale yellow, macroscopic domains with different links ratios should have distinct colors. Secondly, the PCN-160-R% crystals were dissected into unequal parts, which have the same color. In addition, the uniform distribution of linkers can be determined by UV-vis. The crystals of PCN-160-R% were dissected into two unequal parts. The smaller parts and the larger parts were collected respectively, and digested for UV-vis spectra analysis. The linker ratio determined by UV-vis spectra are almost the same for each part (33% for larger parts and 30% for smaller parts). If there is a higher exchange ratio on the surface, a different link ratio would be expected.

3. Scheme 1 is a bit misleading since it implies that terephthalate is used as a ligand when this is not the case.

Response: Thank you pointing it out. We have modified Scheme 1 in the revised manuscript.

4. It is good that the authors are not focused just upon pore size but also upon what can be done with larger pore size.

Response: Thank you. We show that the large pores created by linker labilization can facilitate the diffusion and adsorption of large molecules, including $[\text{Ni}_4(\text{H}_2\text{O})_2(\text{PW}_9\text{O}_{34})_2]^{10-}$ (polyoxometalate or POM, 1.1×1.4 nm), and $\text{Cu}_{24}(\text{BDC-OH})_{24}(\text{H}_2\text{O})_{24}$ (metal-organic polyhedron or MOP, 3.1 nm). In addition, the accessibility of incorporated catalytically active species such as enzyme can be enhanced by linker labilization. These results indicate that the hierarchical porous architectures created by linker labilization have potential applications in adsorption, separation, and catalysis.

5. There are an awfully large number of self-citations. The post-doc who probably wrote the paper should do a better job at citing earlier literature.

Response: Thank you very much for your comments. We have improved the literature citation in the revised manuscript. The following literatures are cited in the revised manuscript.

“Liu, C. *et al.* Establishing porosity gradients within metal–organic frameworks using partial postsynthetic ligand exchange. *J. Am. Chem. Soc.* **138**, 12045-12048 (2016).

Gao, W.-Y., Thiounn, T., Wojtas, L., Chen, Y.-S. & Ma, S. Two highly porous single-crystalline zirconium-based metal-organic frameworks. *Sci. China. Chem.* **59**, 980-983 (2016).

Lykourinou, V. *et al.* Immobilization of mp-11 into a mesoporous metal–organic framework, MP-11@mesoMOF: A new platform for enzymatic catalysis. *J. Am. Chem. Soc.* **133**, 10382-10385 (2011).

Chen, Y. & Ma, S. Biomimetic catalysis of metal-organic frameworks. *Dalton Trans.* **45**, 9744-9753 (2016).

Gkaniatsou, E. *et al.* Metal-organic frameworks: A novel host platform for enzymatic catalysis and detection. *Mater. Horiz.* **4**, 55-63 (2017).

Chen, Y., Lykourinou, V., Hoang, T., Ming, L.-J. & Ma, S. Size-selective biocatalysis of myoglobin immobilized into a mesoporous metal–organic framework with hierarchical pore sizes. *Inorg. Chem.* **51**, 9156-9158 (2012).”

Reviewer #2:

1. The quality of TEM images are not quite good; if possible, the authors should collect some high-resolution TEM images to present more clearly direct evidence for the presence of mesopores in the MOFs. As hinted from the SEM images, the presence of pores larger than 10 nm in some MOFs herein is likely due to the voids of some decomposed MOF particles similar to mesoMOF-1 reported from this research group, which however is not cited (J. Am. Chem. Soc., 2006, 128, 16474-16475). The authors should be cautious about the statement “The maximum pore size can be tuned from 1.5 nm to 18 nm”.

Response: Thank you for your comments and suggestions. We believe that the mesopores observed by N₂ sorption experiments are attributed to the space inside the MOF particles. As shown in the microscopic images of MOF particles, the shape and size of MOF particles were well maintained after linker exchange and acid etching (Fig. 1c and Supplementary Fig. 3). No obvious decomposition of MOF particles were observed. PXRD patterns after solvent removal further prove the maintained crystallinity of MOF samples (Supplementary Fig. 5). Therefore, the mesopores are attributed to the space inside the MOF particles. Linker labilization method partially dissolved MOF particles from inside out, creating large mesopores inside the particle while leaving the particle size visually intact. We collected high-resolution TEM images to prove the presence of mesopores. As shown in the figure below, a sponge like morphology is created after acid etching, corresponding to the mesopores observed by N₂ sorption isotherms. The magnification was set to 71 K to clearly show the mesopores. Further increasing of magnification do not provide more detailed structural information, possibly because the highly porous MOFs tend to be destroyed by stronger electron beams.

The maximum pore size is determined by N_2 sorption isotherms in this work. The definition of maximum pore size could be different depending on the analytic techniques. The following statement has been modified in the revised manuscript.

“The maximum pore size determined by N_2 sorption analysis can be tuned from 1.5 nm to 18 nm by judicious control of the pro-labile-linker content during the MOF synthesis and the acid concentration during the linker labilization process.”

“The maximum pore size by N_2 sorption measurements is tuned from 1.5 nm to 18 nm by judicious control of pro-labile-linker content and acid concentration.”

2. The unit for “Maximum pore size” in the caption of Table S3 should be “angstrom” but NOT “nm”.

Response: Thank you for pointing it out. We have corrected the mistake in the revised manuscript.

3. To better quantify the amount of defects in the MOF, the authors could conduct the titration experiments by referring to some works published by Hupp and Wei Zhou.

Response: Thank you very much for your suggestions. We tried to carry out titration experiments to quantify the amount of defects in the MOF. However, the CBAB linkers dissociates under acidic or basic conditions, generating 4-amino benzoic acid and 4-formylbenzoic acid. The amino group reacts with acid which disturbs the titration

experiments. In addition, some acetates may coordinate to the defect sites of metal clusters during acetic acid treatment, which also affect the accuracy of the titration experiments. As a result, titration experiments cannot provide valid data for this system.

4. The literature citation needs certain improvement. Some references are not quite related, e.g. Ref. 8, 9, 13, 30. Some citations are similar, e.g. Ref. 3&4, Ref. 5&6, and Ref. 37, 38&39 (the year in Ref. 4 and Ref. 6 should be 2012). Ref. 58 should be *J. Am. Chem. Soc.* 2016, 138, 8052–8055. Some other related literatures are missing: e.g. some recent papers regarding the pore size expansion in MOFs (*J. Am. Chem. Soc.* 2016, 138, 12045–12048; *J. Am. Chem. Soc.* 2015, 137, 10508–10511); one more paper regarding the synthesis of Zr-AZDC MOF (*Sci. China Chem.* 2016, 59, 980–983); the first work advocating the benefits of employing hierarchically porous MOFs for enzyme immobilization (*J. Am. Chem. Soc.* 2011, 133, 10382–10385); the first work about the size-selective biocatalysis utilizing hierarchically porous MOF (*Inorg. Chem.* 2012, 51, 9156–9158); some review articles regarding the enzyme immobilization in MOFs (*Mater. Horiz.* 2017, DOI: 10.1039/C6MH00312E; *Dalton Trans.* 2016, 45, 9744–9753).

Response: Thank you very much for your literature information. We have improved the literature citation in the revised manuscript accordingly.

REVIEWERS' COMMENTS:

Reviewer #2 (Remarks to the Author):

The authors have addressed all the comments from the reviewer and not further revision is needed.